# Clinical and Molecular Landscape of ALS Patients with *SOD1* Mutations: Novel Pathogenic Variants and Novel Phenotypes. A Single ALS Center Study

**DOI:** 10.3390/ijms21186807

**Published:** 2020-09-16

**Authors:** Emilien Bernard, Antoine Pegat, Juliette Svahn, Françoise Bouhour, Pascal Leblanc, Stéphanie Millecamps, Stéphane Thobois, Claire Guissart, Serge Lumbroso, Kevin Mouzat

**Affiliations:** 1Centre SLA de Lyon, Hôpital Neurologique P. Wertheimer, Hospices Civils de Lyon, Université de Lyon, 59 Boulevard Pinel, 69677 Bron CEDEX, France; antoine.pegat@chu-lyon.fr (A.P.); juliette.svahn@chu-lyon.fr (J.S.); francoise.bouhour@chu-lyon.fr (F.B.); stephane.thobois@chu-lyon.fr (S.T.); 2Institut NeuroMyoGène, CNRS UMR5310, INSERM U1217, Faculté de Médecine Rockefeller, Université Claude Bernard Lyon I, 8 Avenue Rockefeller, 69373 Lyon CEDEX 08, France; pascal.leblanc@univ-lyon1.fr; 3Institut du Cerveau, ICM, Inserm U1127, CNRS UMR7225, Sorbonne Université, Hôpital Pitié-Salpêtrière, 75646 Paris, France; stephanie.millecamps@icm-institute.org; 4Faculté de Médecine Lyon Sud Charles Merieux, Université Claude Bernard Lyon 1, 69373 Lyon, France; 5CNRS, Institut des Sciences Cognitives Marc Jeannerod, UMR 5229, 69675 Bron, France; 6Laboratoire de Biochimie et Biologie Moleculaire, CHU Nimes, Nimes, Motoneuron Disease: Pathophysiology and Therapy, INM, University Montpellier, 30029 Nîmes CEDEX 9, France; Claire.GUISSART@chu-nimes.fr (C.G.); serge.lumbroso@umontpellier.fr (S.L.); kevin.mouzat@umontpellier.fr (K.M.)

**Keywords:** amyotrophic lateral sclerosis, copper zinc superoxide dismutase 1, *SOD1*

## Abstract

Mutations in the copper zinc superoxide dismutase 1 (*SOD1*) gene are the second most frequent cause of familial amyotrophic lateral sclerosis (ALS). Nearly 200 mutations of this gene have been described so far. We report all *SOD1* pathogenic variants identified in patients followed in the single ALS center of Lyon, France, between 2010 and 2020. Twelve patients from 11 unrelated families are described, including two families with the not yet described *H81Y* and *D126N* mutations. Splice site mutations were detected in two families. We discuss implications concerning genetic screening of *SOD1* gene in familial and sporadic ALS.

## 1. Introduction

Amyotrophic lateral sclerosis (ALS) is a neurodegenerative disorder characterized by degeneration of upper and lower motoneurons leading to death in a median time of three years after onset [1]. Familial ALS (FALS) accounts for almost 10% of ALS cases with more than 20 causative genes described so far [2]. Mutations of copper zinc superoxide dismutase 1 (*SOD1*) gene, the first described, are responsible for 10–20% of FALS cases [3], and more than 200 pathogenic mutations of *SOD1* are reported in the Human Gene Mutation Database (hgmd.cf.ac.uk). The *SOD1* gene is therefore the second gene most frequently linked to ALS, after intronic expansion in *C9ORF72* [4]. Inheritance is autosomal dominant except for the D91A mutation which causes ALS when transmitted in a recessive manner [5]. De novo mutations of *SOD1* can also occur [6]. From a clinical point of view, *SOD1* ALS is characterized by a wide range of phenotypic presentations, from very aggressive [7] to slowly evolving forms [8]. However, to date, only one series has described *SOD1* ALS patients in France [9]. Given that disease-modifying drugs are currently emerging to treat this rare genetic disease, a better understanding of the clinical presentation and evolution of *SOD1* ALS is crucial to enhance prognosis stratification in upcoming clinical trials [10,11].

In this article, we describe the clinical and molecular features of 12 ALS cases associated with *SOD1* mutations and extend the spectrum of genetic variants with two novel ones: *H81Y* and *D126N*.

## 2. Results

### 2.1. Description of the Entire Cohort Features

During the inclusion period, among the 1213 ALS patients newly diagnosed in the Lyon ALS center, 176 patients were tested for *SOD1* mutations (i.e., 100% (80/80) of FALS and 8.5% (96/1133) of SALS. We identified 12 index ALS cases (10 FALS and 2 SALS) with *SOD1* mutations (1% of the whole cohort; 7% of all tested patients; 2% of tested SALS; 12.5% of FALS) belonging to 11 unrelated families, diagnosed by Sanger sequencing (SS) (8/145) or next generation sequencing (NGS) (4/31). No mutations in other ALS genes were found in these 12 patients. The pedigree analysis identified 21 additional related individuals with ALS (no reliable clinical data available). Two patients (family 5 and 7) had no family history of ALS. One family originated from Senegal (family 1), one from Romania (family 9), and the others from France.

The median age at disease onset was 56 years (range 36–72) and the male/female ratio was 0.7/1. Onset of symptoms was spinal in all but one patient (family 9). Bulbar involvement was noted during disease course in three patients (families 6 and 7 and the daughter from family 8). Median disease duration until death or tracheostomy was two years (range 0.5–15). Clinical features of these patients are summarized in Table 1. Change over time in ALS Functional Rating Scale - Revised ALSFRS-R total score since disease onset is depicted in Figure 1.

### 2.2. Molecular Analysis and Clinical Description of Carriers of the Novel D126N and H81Y SOD1 Mutations

Among the 12 *SOD1* mutated patients, three cases belonging to two families displayed previously unknown *SOD1* mutations: *H81Y* for patient II-1 (family 5) and *D126N* for patients II-2 and III-1 from family 8 (Figure 2). Given that the *H81Y* mutation was detected in an apparent sporadic case and to rule out a de novo mutation, we screened both parents of the index case, revealing that the father (patient I-1) was also harboring the same *H81Y* mutation and was asymptomatic at the age of 66. The *D126N* gene co-segregated with the disease in both family members from family 8. In silico analysis of pathogenicity revealed that both variants were considered deleterious according to five software packages (Table 2 (A)). Histidine 81 and aspartate 126 were conserved in all Eukaryota and up to Bacteria and Archaea. According to American College of Medical Genetics (ACMG) criteria, *H81Y* and *D126N* variants were both considered likely pathogenic (class 4) (Table 2 (B)).

#### 2.2.1. *H81Y* Mutation Family

Patient II-1 from family 5 (*H81Y* mutation) was a 42-year-old woman, the oldest of five siblings. Her parents did not present any neurological disorders. She reported a very rapidly evolving weakness of the lower limbs, associated with a 9 kg weight loss, and sphincter disturbances (urgencies without urinary infection), confining her to a wheelchair in only four months. Clinical examination in August 2018, three months after disease onset, revealed a severe flaccid paraparesis (Medical Research Council (MRC) 1–2/5 in distal and proximal lower limb muscles), with predominant calf weakness. Deep tendon reflexes were abolished in lower limbs but preserved in upper limbs without Hoffman’s sign. No fasciculations were initially observed. Needle electromyography showed chronic and active denervation in proximal and distal muscles of lumbar regions, with normal sensory conductions. Motor evoked potential did not show evidence of upper motor neuron involvement. Lumbar MRI showed a slight gadolinium enhancement of lumbosacral nerve roots without thickening. CSF study revealed two white blood cells/mm3 and a slight hyperproteinorachia at 0.54 g/L, without oligoclonal bands. Bone marrow aspiration ruled out the presence of hematological malignancy but showed polyclonal reactive T cells. FDG PET CT was normal. No antiganglioside or onconeural antibodies were detected. Considering the differential diagnosis of dysimmune lower motor neuron disease, she received three courses of IV cyclophosphamid (1g/month during three months) associated with IV corticosteroid bolus (1g/month during three months) without any efficacy. In March 2019, weakness associated with fasciculations had spread rapidly to both arms and, from May 2019, use of noninvasive ventilation (NIV) was mandatory due to severe dyspnea. She died in September 2019 from respiratory distress 16 months after onset.

#### 2.2.2. *D126* Mutation Family

The father (II-2) and his daughter (III-1) from family 8 (*D126N* mutation) both developed first symptoms in summer 2017 and were simultaneously diagnosed from suffering of ALS in October 2017. Family history revealed that patient II-2’s father (I-1) died from ALS at 85 years old. Patient II-2 was a 62-year-old man reporting a five-month history of progressive weakness in the left thigh, without sensory disturbance or pain. His only personal history was the occurrence, since February 2017, of acute intermittent headaches, lasting more than three hours, more frequent at nights, and associated with a right mild ptosis and pupillary miosis which persisted between episodes (Horner’s sign), without tearing of the eye or congestion. Cervical CT scan ruled out a carotid dissection and he was treated with sumatriptan with partial efficacy. Neurological examination revealed muscle weakness of the left lower limb with proximal predominance (MRC 3/5 for quadriceps and 4/5 for tibialis anterior). Deep tendon reflexes were diminished in four limbs and the left patellar tendon reflex was absent. Fasciculations were found in both lower limbs. No upper motor neuron signs were present. Neurophysiological examination showed acute and chronic denervation, with multi-metameric distribution and fasciculations in the lower limbs, predominantly in the left limb, without upper limb abnormalities. Fourteen months after onset, he could not walk or stand. NIV was started in February 2019 and he died from acute respiratory failure in September 2019, 11 months after onset.

His daughter (III-1), a 42-year-old woman was referred for a weakness in her right hand lasting for three months. Neurological examination in October 2017 showed motor deficit and atrophy in the right upper limb with MRC 2/5 in distal and 3–4/5 in proximal muscles. Sensory examination was normal. Deep tendon reflexes were diminished and no upper motor neuron signs were present. Fasciculations were observed in four limbs. Needle electromyography showed acute and chronic denervation in the cervical and lumbar regions. The disease progressed rapidly in the right upper and lower hemiface inducing a Bell’s phenomenon which progressed in the same way to the contralateral hemiface, in association with dysarthria, dysphagia, and tongue fasciculations. Due to respiratory failure and dysphagia, tracheostomy and gastrostomy were inserted in April 2018. She progressively developed complete flaccid tetraplegia. Cognitive functions were preserved at least until April 2020 when, despite ocular protection, she developed severe blindness due to bilateral infectious keratitis, leading her to a locked-in state. She was still alive at the time of writing of this manuscript.

## 3. Discussion

The present study describes the clinical spectrum of ALS patients harboring *SOD1* mutations diagnosed over a decade in the ALS center of Lyon, France. This is the second largest series of *SOD1* ALS patients reported in France so far [9]. The reported prevalence of *SOD1* mutations in FALS (12.5%) and in tested SALS (2%) is in the range of previous studies in European populations in which rates vary from 2.6 to 14% for FALS, and 0.63–1.64% for SALS [15]. The global prevalence of *SOD1* mutations was 7% (12/176) in tested patients. By comparison, 378 patients with ALS were tested for *C9ORF72* in our institution between 2010 and 2020. Among them, 65 had the intronic expansion, which corresponded to a global prevalence of 17%, combining all categories (FALS and SALS). To note, given the incidence of ALS in France [16] and the total number of ALS patients followed in our center, the 12 patients reported in the present study could represent approximatively 6% of all *SOD1* cases diagnosed in the country during the same period.

Overall, except for the homozygous *D91A* mutation, known to display a slower course [5,11], most of the mutations were associated with an aggressive evolution: two thirds of patients presented respiratory insufficiency and the median survival time was only two years, which is shorter than the median disease duration of three years previously reported in France [9], but in line with the largest *SOD1* cohort to date [11]. Inheritance was dominant in all cases except for the homozygous *D91A* mutation, which is recessive. Except for family 5 (see below), penetrance of the disease seemed complete in all families with dominant inheritance (data not shown), although the lack of reliable data from their relatives precluded the possibility to assess phenotypic variability among most families.

Herein, we report a novel *H81Y* mutation. Mutation of the same amino acid 81 (*H81R*) has been reported once by Alexander et al., in 2002 [6], who described the same core features: young age at onset, lower limb onset with lower motor neuron predominance, and aggressive course. Initial diagnosis of dysimmune lower motor neuron disease was suspected in both cases and both were treated with cyclophosphamide, without success. The patient herein did not present liver abnormalities as reported by Alexander et al., possibly due to the abundance of mutated *H81R SOD1* in this organ. However, the presence of reactive polyclonal T cell in the bone marrow of the patient described here might be explained by the same mechanism of protein overload. We also found, in this particular case, gadolinium enhancement without thickening of lumbosacral nerves, a very rare feature in ALS, only described in SALS [17,18]. The 81 position is situated in one of the four amino acid zinc ligands of *SOD1*. This strategic position was predicted to induce an aggressive phenotype [19] which was corroborated by the present report. However, surprisingly, the father of the index case also harboring the *H81Y* mutation was still asymptomatic at 66 years old, indicating variable expressivity of the disease or incomplete penetrance of this variant. To date, molecular mechanisms underlying age at onset or penetrance are still poorly understood. Although we cannot exclude the possibility that the father of the index case is pre-symptomatic, further investigations will be interesting to understand the link between this variant and ALS pathophysiology.

We identified a second novel mutation, *D126N*. Two different mutations of amino acid 126 have already been reported, but clinical data are sparse for *D126H* [20] and absent for *D126A* [21]. The two patients from family 8 both presented an aggressive spinal-onset ALS with pure lower motor neuron involvement. Furthermore, the father also presented an unusual association with a painful Horner’s sign, also called Raeder’s syndrome. The persistence of Horner’s sign independently of headaches episodes was not consistent with cluster headache. Given that *SOD1* mutations can affect the autonomic system [22] involved in the pathogenesis of Raeder’s syndrome [23], this very rare clinical presentation, starting simultaneously with the motor signs, could represent a specific clinical feature associated with the *D126N* mutation. However, the absence of pathological examination of this patient concerning the presence of *SOD1* aggregates in the sympathetic system, and no report of such associations with ALS, does not allow us to draw definite conclusions regarding a possible causal link. Another atypical feature in this family was the presence of bilateral peripheral facial diplegia in the daughter. However, such a manifestation has already been reported with *I114T* [24] and *A5V* [25] mutations. Age of onset in the three patients from this family was heterogeneous (42, 65, and 85 years-old). Although the reason for this variability is unknown, clinical heterogeneity has been described in many autosomal dominantly inherited diseases including in several *SOD1* mutations [26].

Besides the description of these two novel variants, we reported the first clinical description of two patients from separate families harboring the intronic c.358-10T>G splice site mutation, originally identified by Sapp et al., in 1995 [14]. Both cases presented the similar phenotype of predominant lower motor neuron signs with rather symmetrical involvement of proximal muscles at onset in upper and lower limbs, a pattern very similar to spinal muscular atrophy (SMA), ruled out by the absence of *SMN1* mutations in both cases. To our knowledge, this SMA-like phenotype in ALS has not been reported before.

Clinical features of both *H44R* patients was relatively similar with a previous report of this mutation [11] describing a classic ALS phenotype, combining upper and motor neuron signs associated with a very aggressive course, and death occurring 1 to 3 years after disease onset.

One of the two patients with the homozygous *D91A* mutation had an unusually slowly evolving phenotype, associating pes cavus, lower limb deep tendon reflexes abolition, and vocal cord involvement without dysarthria. Although described with *A102T*, *I150T*, *A5V*, *I114P* and *G148S SOD1* variants [27], vocal cord paralysis has not been previously reported with homozygous *D91A* mutation. Dysphonia was also a prominent clinical feature in the patients harboring the *G148D* and *P67S* mutations.

The *C7S* mutation has been reported in one family from African American ancestry [12]. We report here a second family originating from Senegal, confirming that *SOD1* mutations can occur in Africa. The non-motor symptoms such as reduced vibration perception and neurogenic bladder described for this mutation have previously been reported with *G94S* [28], *L107V* [29], and *V119L* [30] mutations. 

## 4. Materials and Methods

### 4.1. Data Collection

All *SOD1* gene mutated ALS patients (*SOD1* ALS) followed in the referral ALS center of Lyon, France, during the 2010–2020 period, were enrolled. For each patient, the following information were collected: age at onset, gender, site for onset of symptoms, disease duration until death or tracheostomy, upper and lower motor neuron signs, presence of respiratory failure, ALSFRS total score, and electrophysiological data. All patients signed an informed consent form concerning use of their data for genetic research purposes. Investigations were carried out following the rules of the Declaration of Helsinki of 1975, revised in 2013.

### 4.2. Genetic Screening

Patients were submitted to genetic tests if they reported family history of ALS, if their phenotype evoked the presence of *SOD1* mutation (e.g., young onset, aggressive course with predominant lower motor neuron involvement, or prominent vocal cord impairment) or if they specifically requested these analyses. Genetic analysis was performed in Nimes University hospital. From 2010 to 2017, *SOD1* mutations were screened by SS only, and from 2017 to 2020, by targeted NGS, performed on an Ion Torrent S5 sequencer (Thermo Fisher Scientific, Waltham, MA, USA). The ALS panel included exons and flanking regions of *ALS2*, *ANG*, *CHCHD10*, *CHMP2B*, *DAO*, *DCTN1*, *DPYSL3*, *ELP3*, *ERB4*, *EWSR1*, *FIG4*, *FUS*, *HNRNPA1*, *HNRNPA2B1*, *LMNB1*, *MATR3*, *NEFH*, *OPTN*, *PFN1*, *PRPH*, *SETX*, *SIGMAR1*, *SOD1*, *SPAST*, *SPG11*, *SQSTM1*, *TAF15*, *TARDBP*, *TBK1*, *TREM2*, *TUBA4A*, *UBQLN2*, *UNC13A*, *VAPB*, and *VCP*. All patients were first screened for abnormal hexanucleotide repeat expansion in *C9ORF72* gene using repeat primed PCR. All *SOD1* mutations detected by NGS were confirmed by SS. The ACMG criteria [31] were used for classifying variants according to likelihood of pathogenicity.

### 4.3. Nomenclature

The Human Genome Variation Society nomenclature [32] was used to name the different *SOD1* variants, i.e., using one amino acid shift compared to past nomenclature. For example, the former *D90A* mutation was named *D91A*.

## 5. Conclusions

The present series added two novels variants (*H81Y* and *D126N*) to the list of known *SOD1* mutations, and thus expanded the spectrum of clinical manifestations associated with *SOD1* mutations. The report of two patients with *SOD1* mutations in an apparent sporadic context (families 5 and 7) highlighted that in the current era of emerging disease-modifying drugs to treat *SOD1* mutated patients, and with the upcoming urgent need to screen all families, only systematic screening of all ALS patients, including apparent sporadic cases, can capture all patients harboring these mutations. We also suggest the need for careful attention in regard to the presence of the intronic c.358-10T>G splice site mutation, which may be more frequent than previously reported, as attested by its high prevalence in this cohort (2/11 families). The polymorphic clinical features associated with *SOD1* mutations also underlined that *SOD1* testing should be performed in all atypical motor neuron diseases not fulfilling Airlie House criteria for ALS diagnosis, especially in case of absence of upper motor neuron signs or with SMA-like presentation, or in association with sensory signs, vocal cord impairment, neurogenic bladder, facial diplegia, or gadolinium root enhancement.

## Figures and Tables

**Figure 1 ijms-21-06807-f001:**
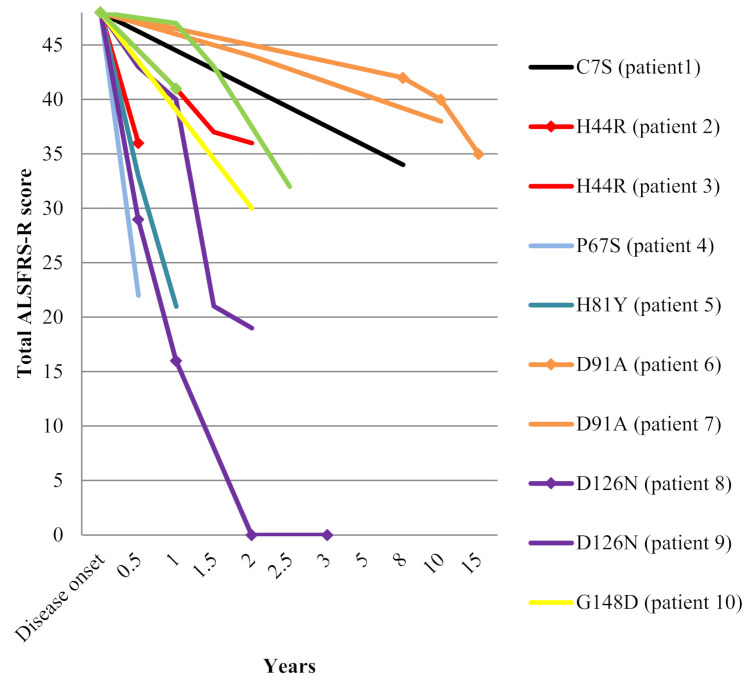
Decline in ALSFRS-R total score among the 12 *SOD1* mutated patients.

**Figure 2 ijms-21-06807-f002:**
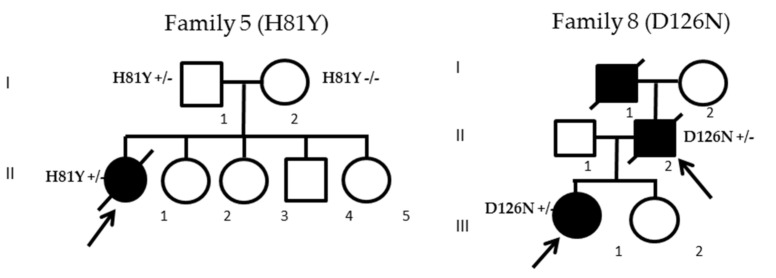
Pedigrees of the two families harboring the *H81Y* and *D126N* mutations. Arrows indicate index patients; black filled icons indicate ALS cases.

**Table 1 ijms-21-06807-t001:** Comparative analysis of clinical manifestations in patients with copper zinc superoxide dismutase gene 1 (*SOD1*) mutation.

Family N°	Variant	First Description of the Mutation	Family History of ALS	Sex	Age at Onset (y)	Site of Onset	Disease Duration (y)	Bulbar Palsy	UMN Signs (UL/LL)	LMN Signs (UL/LL)	Respiratory Failure	Additional Features
1	C7S +/−	[12]	+	M	39	LL	10	−	+/−	+/+	+	Neurogenic bladder Reduced vibration perception No sensory neuropathy/Abnormal SEP
2	H44R +/−	[3]	+	M	69	LL	3	−	+/+	+/+	+	Sensory neuropathy
3	H44R +/−		+	F	44	UL	1	−	+/+	+/+	+	-
4	P67S +/−	[13]	+	M	72	LL	0.5	−	−/−	+/+	+	Vocal cord involvement
5	H81Y+/−	present study	−	F	42	LL	1	−	−/−	+/+	+	Neurogenic bladder Gadolinium enhancement of lumbosacral roots Reactive T cells in bone marrow
6	D91A +/+	[5]	+	F	58	LL	>15	+	+/+	−/−	−	Proprioceptive ataxia No sensory neuropathy/Abnormal SEP
7	D91A +/+		−	F	36	LL	>6	+	+/+	+/+	−	Vocal cord involvement Pes cavus
8	D126N +/−	present study	+	M	65	LL	2	−	−/−	+/+	+	Painful Horner’s sign (Raeder syndrome)
	D126N +/−		+	F	42	UL	>3 (tracheostomy after 1)	+	−/−	+/+	+	Facial diplegia with bell’s phenomenon
9	G148D +/−	[9]	+	F	59	bulbar	2	+	+/+	−/−	+	Vocal cord involvement
10	c.358-10T>G +/−	[14]	+	F	54	LL	1 (sudden death)	−	−/−	+/+	−	Proximal lower and upper limb weakness (absence of *SMN1* mutation)
11	c.358-10T>G +/−		+	M	67	LL	>3	−	−/−	+/+	−	Spinal muscular atrophy-like pattern (proximal > distal) of muscle atrophy (absence of *SMN1* mutation) Sensory neuropathy Reduced vibration perception

Key: +: present; −: absent; (−): absent at onset; +/−: heterozygous state; +/+ homozygous state; F: female; LL: lower limb; LMN: lower motor neuron involvement; M: male; UMN: upper motor neuron involvement; UL: upper limb; SEP: somatosensory evoked potentials.

**Table 2 ijms-21-06807-t002:** (**A**): In silico pathogenicity prediction tests for *H81Y* and *D126N SOD1* mutations. Human genome variation society (HGVS) nomenclature V2.0 according to mRNA reference sequence GenBank: NM_000454.4. Nucleotide numbering uses +1 as the A of the ATG translation initiation codon in the reference sequence, with the initiation codon as codon 1. (**B**): Evidence of pathogenicity and classification according to ACMG criteria.

	Mutations	SIFT	PolyPhen-2	CADD Phred Score	Mutation Taster	Panther	MAF in GnomAD	dbSNP
(**A**)	c.241C>T, (H81Y)	Deleterious	Probably damaging	28.0 (Deleterious)	Disease causing	Probably damaging	Absent	Absent
	c.376G>A, (D126N)	Deleterious	Probably damaging	32.0 (Deleterious)	Disease causing	Probably damaging	Absent	Absent
		**Evidence of Pathogenicity**						**Classification of Sequence Variants**
(**B**)		**(MODERATE)** **Located in a Mutational Hot Spot and/or Critical and Well-Established Functional Domain without benign Variation**	**(MODERATE)** **Absent from Controls**	**(MODERATE)** **Novel Missense Change at an Amino Acid Residue Where a Different Missense Change Determined to be Pathogenic Has Been Seen Before**	**(SUPPORTING)** **Co-segregation with Disease in Multiple Affected Family Members**	**(SUPPORTING)** **Missense Variant in a Gene that has a Low Rate of Benign Missense Variation and in which Missense Variants are a Common Mechanism of Disease**	**(SUPPORTING)** **Multiple Lines of Computational Evidence Support a Deleterious Effect on the Gene or Gene Product**	
**Mutations**	c.241C>T, (H81Y)	Yes	Yes, MAF = 0	Yes	No	Yes	Yes	Likely Pathogenic
	c.376G>A, (D126N)	Yes	Yes, MAF = 0	Yes	Yes	Yes	Yes	Likely Pathogenic

Abbreviations: CADD = combined annotation-dependent depletion; dbSNP = single nucleotide polymorphism database; MAF = minor allele frequency; five software packages investigating in silico pathogenicity were used to classify variants from benign (tolerated) to disease causing (deleterious, probably damaging). A variant with a CADD score >20 indicates that it is among the 1% most deleterious variants in the genome and >30 among the 0.1%. MAF in control populations and dbSNP identification number are also indicated.

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
