# Peer review of "Clinical and Molecular Landscape of ALS Patients with SOD1 Mutations: Novel Pathogenic Variants and Novel Phenotypes. A Single ALS Center Study"

_ijms, 2020, doi:10.3390/ijms21186807_

Round 1
Reviewer 1 Report
In this resubmission, Bernard and colleagues improved their previous presentation of this same data.
As in that previous occasion, the interest of this work is limited as the clinical features related to the mutations they identify are within the normal range of what can be expected for ALS. But even so, this data deserves being known by the community.
What I find missing from this report is:
- A clear explanation of the criteria used to analyze SOD1 in ALS cases, specially in sporadic cases.
- Clinical general information on the population analyzed: similarities and differences between SOD1 mutations and no mutations.
- If other mutations appear in their series (C9ORF72, for instance), similarities and differences (if any) depending on the gene with mutations.
- Which/How many samples are screened for SOD1 vs samples screened with the NGS panel.
- How they define the FALS group (specially taking into account that "family 5" only shows one apparently sporadic case)
- Similarly, patient from family 7 does not show family history. Why is she considered a familial case?
- Is there history of other neurodegenerative disorders in any of those families? I am specially thinking in frontotemporal dementia.
- It would be very interesting to have a segregation analysis of family 5 with 4 at risk individuals.
- I still find missing some comparison with other series not only from France but also from other populations.
Surprisingly, the authors state (pg 4, line 69/70) that "...penetrance seems complete..." which is curious taking into account the relatively low amount of data on other possible carriers of the disease and the fact that, at least in family 5, there is an unaffected carrier with more than 65 years of age. Obviously he could develop ALS later in life.
Author Response
Reviewer 1
In this resubmission, Bernard and colleagues improved their previous presentation of this same data.
As in that previous occasion, the interest of this work is limited as the clinical features related to the mutations they identify are within the normal range of what can be expected for ALS. But even so, this data deserves being known by the community.
What I find missing from this report is:
- A clear explanation of the criteria used to analyze SOD1 in ALS cases, especially in sporadic cases.
We added this sentence in the main text:
“Patients were submitted to genetic tests if they reported family history of ALS, if their phenotype evoked the presence of SOD1 mutation (e.g. young onset, aggressive course with predominant lower motor neuron involvement or prominent vocal cord impairment) or if they specifically requested these analyses.”
- Clinical general information on the population analyzed: similarities and differences between SOD1 mutations and no mutations.
176 patients were screened for SOD1 mutations and we regret we cannot obtain with a reasonable time these data.
- If other mutations appear in their series (C9ORF72, for instance), similarities and differences (if any) depending on the gene with mutations.
The objective of this article was not to compare the SOD1 population with FALS associated this C9ORF72, FUS and TARBP mutations followed in our institution.
We regret that we cannot obtain these data with a reasonable time.
- Which/How many samples are screened for SOD1 vs samples screened with the NGS panel.
We added this sentence in the main text:
“We identified 12 index ALS cases…….belonging to 11 unrelated families, diagnosed by Sanger sequencing (SS) (8/145) or Next Generation Sequencing (NGS) (4/31).
- How they define the FALS group (specially taking into account that "family 5" only shows one apparently sporadic case)
There is currently no consensus on the definition of familial ALS (FALS).
However, ALS is generally accepted to be familial if one or more first- or second-degree relatives are reported to suffer from the condition (Valdmanis P, Rouleau G. Genetics of familial amyotrophic lateral sclerosis. Neurol. 2008;70:144 – 52.).
The two patients from family 5 and 7 are therefore considered sporadic cases, as attested page 2 line 55 : “We identified 12 index ALS cases (10 FALS and 2 SALS) with SOD1 mutations”,
To note, other criteria exist: Byrne et al. in 2011 (Amyotrophic Lateral Sclerosis, 2011; 12: 157–159) considered “possible FALS” a “Sporadic ALS patient with no family history, but positive for a FALS gene” but we did not use this definition.
- Similarly, patient from family 7 does not show family history. Why is she considered a familial case?
See precedent answer : she is considered sporadic.
- Is there history of other neurodegenerative disorders in any of those families? I am specially thinking in frontotemporal dementia.
Although we agree that frontotemporal dementia can unfrequently occur in association with SOD1 mutations, we did not report such association in this cohort.
- It would be very interesting to have a segregation analysis of family 5 with 4 at risk individuals.
Unfortunately, segregation analysis could not be obtained for the 4 at risk individuals. However, given that they all are asymptomatic and younger than the index case, the presence or absence of the pathogenic variant could not help to determine the penetrance for the disease.
- I still find missing some comparison with other series not only from France but also from other populations.
We add in the main text : “….previously reported in France [9] but in line with the largest SOD1 cohort to date [11].”
Surprisingly, the authors state (pg 4, line 69/70) that "...penetrance seems complete..." which is curious taking into account the relatively low amount of data on other possible carriers of the disease and the fact that, at least in family 5, there is an unaffected carrier with more than 65 years of age. Obviously he could develop ALS later in life.
We added this sentence in the main text: “Except for family 5 (see below), penetrance seems complete…..”
Reviewer 2 Report
The authors responded to all my comments.
Author Response
thank you very much
Round 2
Reviewer 1 Report
In this new submission, the authors appropriately answered most of my previous comments. Unfortunately, they seem unable to answer to questions #2 and #3 which I consider will certainly enrich their message and make their work more interesting. On the other hand, I am puzzled by the fact that an authorship list with a number of clinicians could have problems to gather all the required information (provided there are not personal conflicts within the research team)
This manuscript is a resubmission of an earlier submission. The following is a list of the peer review reports and author responses from that submission.
Round 1
Reviewer 1 Report
In this manuscript, Bernard and colleagues report their experience in ALS clinical management for the last 10 years focusing in patients showing SOD1 mutations. As part of this, they report two new mutations and some clinical features on other SOD1 mutated individuals with ALS.
Moreover, the authors include some very summarized information on published material about other clinically relevant findings in other patients with the same mutation as those studied by them.
Several aspects on their work are still "open" in their manuscript:
- The difference in sex ratio compared to most published work that shows how men are more frequently suffering ALS than women (for example see: Liu et al Brain Res 1693A:121-6, 2018).
- Also, the authors do not include any information on evolutionary conservation of the sites mutated in their sample nor on an apparent anticipation that might be occurring in the pedigree with the p.D126N mutation.
In addition, there is an apparent inconsistency when talking about the p.H81Y mutation, if the mutation is predicted to cause an "aggressive phenotype" how to explain that, at the same time, the mutation appears to show reduced penetrance or expressivity?
Minor comments:
- Authors must include in their manuscript the reference of the RefSeq gene (or, in general, the entry ID) that they use for their numbering.
- Authors should use either one- or three-letter code for amino acids, but not both.
- In table one, when referring to the p.D126N mutation, include the actual MAF in gnomAD for that particular mutation instead of "yes".
Reviewer 2 Report
The authors identified multiple SOD1 gene mutations in familial and sporadic ALS cases of their cohort, including two newly identified point mutations and a splice site mutation. Characteristic symptoms in each variant are noted in contrast to the previous reports, showing that a similar tendency is seen in the French cohort. Also, they present the clinical manifestations of ALS due to the newly identified SOD1 mutations, and give important guidance in determining the need for SOD1 genetic testing. However, the following points should be clarified.
- How many cases are found to have intronic expansion in C9ORF72 in this study? The expansion mutation of the gene should be the most frequent in European ALS, so it is important to show that compared to the frequency of SOD1 mutation.
- Why do the patients in Family 8 have such a discrepancy in the age of onset and symptoms? Can the authors discuss that referring to other reports?
- What is the basis for diagnosing Raeder's syndrome with headache and autonomic symptoms of the patient II-2 in Family 8? What is the difference from cluster headache, which is often misdiagnosed as Raeder's syndrome?